# Investigation of Community Energy Business Models from an Institutional Perspective: Intermediaries and Policy Instruments in Selected Cases of Developing and Developed Countries

Naimeh Mohammadi 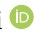

Department of Work, Technology and Participation, Technische Universität Berlin, 10587 Berlin, Germany; naimeh.mohammadi@win.tu-berlin.de

**Abstract:** Community energy development and the empowerment of customers as producers are the main contributors to decentralized market solutions in energy transition policy. Despite the growing literature on community energy projects from the perspectives of various business models, drivers, and barriers, few studies display the impact of institutional factors on the community energy business model configuration. Using insights from Ostrom's institutional framework, this study develops a conceptual framework comprising policy instruments and the intermediaries that configure the various community energy business models, and it examines this framework in the developed world of northwestern European countries (Germany, Denmark, Belgium, and the UK) and in selected cases in developing countries (Rural Central America, South Africa, Iran, and Indonesia). The findings indicate that ambitious renewable energy consumption targets and national policies in northern EU countries have resulted in political and financial incentives, as well as greater financial security than other investment markets, which encourage citizens to contribute to the proliferation of community energy. On the other hand, in the studied developing countries, top-down energy policies and a centralized energy system are insufficient for participatory energy planning. Due to unsupportive policies, a lack of appropriate regulatory frameworks, and a lack of institutional support in these countries, the initiation of community energy projects requires the presence of intermediaries such as developers who work 'in-between' other actors, such as energy providers, users, or regulators.

**Keywords:** energy transition; community energy; business models; institutional environment; intermediaries; policy instruments; European nations; developing countries

## 1. Introduction

Community energy (CE) projects are often "grassroots" initiatives and are governed and managed by the local community [1].

Positive attitudes toward alternative energy sources and environmental issues, as well as socio-institutional characteristics such as the normative spirit of communication, resilience, and the awareness of technical aspects of energy systems, are critical for CE's success [2–4]. Furthermore, the involvement of community members is crucial as they can be active participants in the transition process. Community members are voluntary active actors from households, public authorities, and social or private sectors that require diverse portfolios of innovative business models and intermediating entities [5].

The key factors that place a greater emphasis on the concept of bottom-up energy generation and the emergence of local CE businesses include the adoption of new technologies and smart meters, as well as advancements in information technology and energy resource distribution [6,7]. These new CE business models are assuming new roles as energy suppliers and service providers, which is a departure from the conventional small

and medium-sized citizen communities that are mainly involved in the generation and use of electricity.

Moreover, by promoting energy efficiency and energy-saving activities, such as building modernization and car sharing, these communities can help to reduce their overall energy consumption and promote sustainable living [8,9].

In developed countries, ambitious targets for renewable energy consumption and emission reduction goals have led to policy supports in terms of financial incentives, loan capital with preferential conditions, and priority to grid access [10]. These policy interventions have provided technological and political legitimacy to innovative communities, particularly those with a long-standing tradition of cooperation [11].

In developing countries, the main drivers of CE projects are unfulfilled basic needs, particularly the provision of electricity and heat in rural and isolated areas. The concept of enablers in developing countries, such as the bottom of the pyramid and microfinance principles aligned with social entrepreneurship, offer promising solutions to address these needs. These enablers can provide access to capital, technology, and knowledge, which are critical for the development and implementation of CE projects in these regions [12].

*Identified Research Gaps and the Aim of the Paper*

The literature on CE has explored different aspects of CE projects, including drivers, barriers, conceptualization, and business models. In the research of Bothelho, the focus is on the role of business models as enablers of the growth of prosumers (i.e., consumers who also produce energy) in the CE market. The study identifies the primary characteristics of market design and the related regulatory requirements that can enable the growth of prosumers in the CE market [13]. In the research of Brummer and Reise, the thematic focus is on the value proposition offered by community projects and the interrelationship between policy schemes and CE development across Europe and the US [5,14].

Intermediary actors play a crucial role in facilitating the development of CE business models by connecting different stakeholders, such as investors, developers, and policymakers, and providing support and expertise to these stakeholders. Nolden's research examines the role of these intermediary actors in facilitating and brokering CE business models in England [15].

The research of Brummer, Reise, and Nolden highlights the importance of intermediary actors in the development of CE BMs. Brummer and Reise's research focuses on the value proposition offered by community projects and the interrelationship between policy schemes and CE development across Europe and the US. They suggest that intermediary actors can help to bridge the gap between policymakers, investors, and developers by providing support and expertise to these stakeholders.

Nolden's research specifically examines the role of intermediary actors in facilitating and brokering CE business models in England. He suggests that these actors can help to overcome the barriers to the adoption of CE business models, such as the lack of knowledge and expertise, access to finance, and regulatory challenges.

The study also explores the drivers of CE business models in England, including policy frameworks, market structures, and technological innovation.

As noted by Neska, the integration of renewable energy sources, the adoption of digital technologies, and the liberalization of energy markets are driving the transformation of smart grids. Neska's research focuses on the conceptual design of energy community market topologies, which are emerging as new models for the deployment of renewable energy resources and the integration of energy markets [16].

The work of Engelken is significant as it highlights the distinction between developing and industrialized countries in terms of the barriers, motivations, and difficulties of the CE BMs [17]. Developing countries often face more significant challenges in this regard due to factors such as limited resources, inadequate infrastructure, and political instability. On the other hand, industrialized countries may face different obstacles related to the regulatory environment or market conditions.

Vallecha and Vargas's research provides insights into the grassroot barriers and enablers of CE in India and Central America, providing insights into the capacity requirements for the governance of CE initiatives in these regions. However, the study does not provide an explanation of the business model mechanisms behind these initiatives [12,18].

In all of the aforementioned research studies, the scholars have debated the barriers and enablers and the relationships between business models and policy support. While previous researchers derived their valuable findings from empirical analyses of existing EC projects, none of them investigated or focused on conceptualizing CE structures from an institutional perspective; in addition, there is an absence of an in-depth debate about the relationships between institutional arrangements (policies, process, and intermediaries) and community energy business models.

This work fills this gap. This research provides an insight into how the emergence of CE initiatives is influenced by policy support instruments and by the role of the intermediary between communities in the social, private, and public sectors [19,20]. The main objective of this study is to investigate the mechanisms through which the actions of intermediaries (both traditional and new) in relation to policy instruments (FiTs, net metering, and pilot projects) interact iteratively to enable the emergence of new CE practices.

Hence, this study proposes a conceptual framework of community energy business models from an institutional perspective (Figure 1); this framework will help to create a better understanding of the mechanisms through which intermediaries and policy instruments interact to enable the emergence of new CE practices.

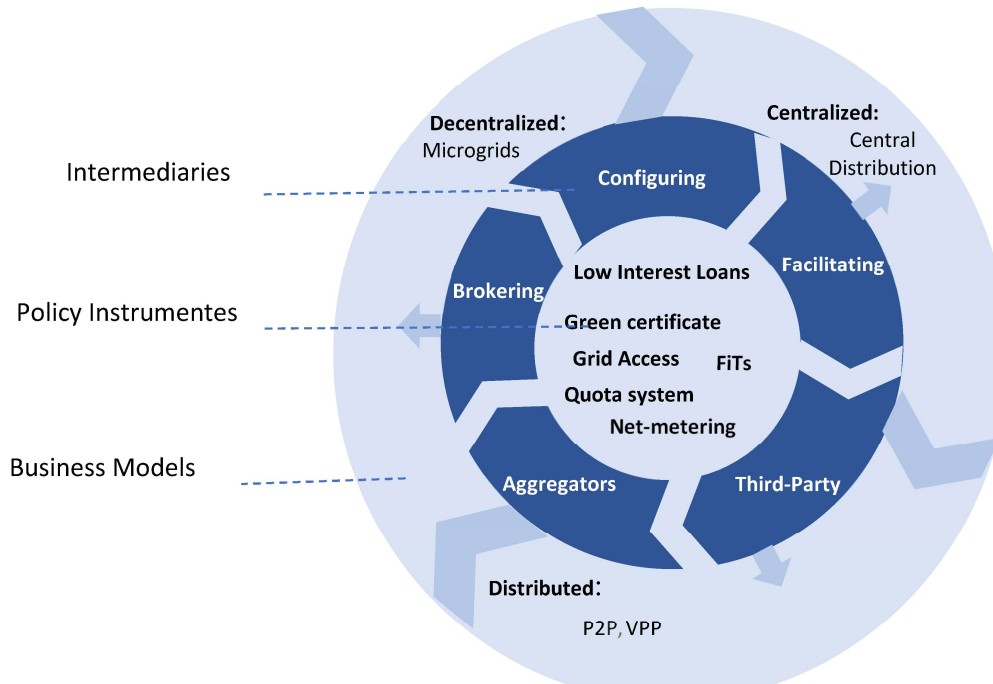

**Figure 1.** Conceptual framework of community energy business models from institutional perspective: intermediaries and supportive policy instruments (proposed by the author).

This research will answer the following research questions:

(1) Which kinds of policy support instruments and intermediary partnerships account for the emergence of CE BMs?
(2) How and what types of BMs were developed for the selected cases?
(3) What is the mechanism that explains sustainable energy communities in selected cases?

For a more accurate comparison of the contextual factors, selected case studies from both developing and developed nations were chosen. The contribution of this research is to

employ an empirical method to compare developing and developed countries in northern European countries, and the aim is to explain the differences in institutional arrangements between these two groups of countries.

The novelty of this work is the interdisciplinary approach to CE structures.

The subject areas of policy sciences, business studies, institutional frameworks, renewable energy sources, and the latest trends in technology were all brought together. The comprehension of how these areas affect each other allows for a comprehensive understanding of the EC concept, its complexity, and the possible direction of its evolution.

Furthermore, the paper juxtaposes theory and practice; it first describes the conceptual institutional framework of community energy, business models, and the role of intermediaries. Then, it presents the practical application of EC projects and their real market designs. This study brings together research from both developing and developed countries and forms a conceptual framework for the business models that have been used to deploy community-based energy projects.

The remainder of the paper is structured as follows:

The second section provides the conceptual framework of the research proposed by the author. The third section discusses the research design and methodology. Section 4 offers the findings of the case studies examined in accordance with the conceptual framework. Section 5 examines prospective solutions, future trends, and evolution in the context of both worlds, and Section 6 concludes the paper.

## 2. Conceptualizing Community Energy Business Models from an Institutional Perspective

Ostrom's institutional analysis and framework development (IAD) facilitates the analysis of the institutional processes through which individual and collective choices occur. In this framework, the action arena in which groups of individuals interact and produce outcomes under the influence of exogenous variables is comprised of all the social, institutional, physical, and contextual factors [21].

Indeed, the configuration of the various CE BMs, as the action arena, is entrenched in various social and structural institutions and includes various supportive policies, intermediaries, and contextual factors [22,23]. In fact, the contextual factors, policy instruments, and intermediaries all contribute to the success and positioning of energy community businesses.

Hence, this research explains the various types of policy support instruments and intermediary partnerships involved in the emergence of CE business models.

Further explanation of the components in the conceptual framework of "community energy business models from an institutional perspective: intermediaries and supportive policy instruments" is provided below, Sections 2.1–2.3

In Section 2.1, the different types of community energy business models are explained. Section 2.2 clarifies the role of the intermediaries, and Section 2.3 specifies the role of the policy instruments.

### 2.1. Different Categorization of CE BMs

A business model, according to the literature, consists of four essential elements: the value proposition, customer relationships, infrastructure management, and financial features. The value proposition concerns the product and the benefits provided to the market by the company. Customer management describes how potential customers are segmented by distribution channels, whereas infrastructure management describes key partners and capability configuration. Finally, the financial component clarifies the cost structures and revenue model [12].

In a CE BM, two elements of production and customer management make important differences. Firstly, traditional electricity production resources are being adapted to renewable sources such as wind power and PV. Secondly, traditional consumers have

transformed into prosumers, who are becoming producers and key partners in electricity generation [24].

Following a review of the business model literature, it appears that the most appropriate classifications for covering all categorization types of CE BMs are based on Gui's research, which classified different types of CE BMs into centralized, decentralized, and distributed BMs [1].

The ability of centralized CE BMs is to easily integrate into the current, largely centralized regime of electricity generation. Cooperatives, trustees, and public companies are some examples [1]. The central feature of a centralized CE BM is its ability to easily merge into the current, largely centralized regime of electricity generation. Some examples include cooperatives, trustees, or public companies [1]. In this model, energy technologies are purchased directly by end users and used for investments in energy generation and storage. Prosumers have been incentivized to act as investors, decision makers, and customers in order to benefit from self-consumption and bill reduction on the basis of longstanding PPA agreements. The key business partners communicate directly with energy suppliers and the distribution system in order to take advantage of remunerative financing terms in their bulk asset acquisition [5,16].

Members of a decentralized community typically belong to a local area, neighborhood, village, or municipality, and they are responsible for managing the electricity production, storage, and consumption in order to achieve self-sufficiency and autonomy from centralized energy systems. They can own the energy resources individually or collectively as a group, as part of community microgrids, or as a part of the integrated CE system [1].

A microgrid is a controlled small-scale power system that can either be connected to the main grid or operated on an isolated grid. Microgrids are designed similarly to distribution networks, with the main difference being the regional control of demand and supply balances. A microgrid can be connected to a point of common coupling (PCC) to sell the excess energy to the grid [25].

A distributed CE BM is a network of households and businesses that generates and distributes electricity independently. This BM could be implemented in smart cities, and it incentivizes the initiation of new start-ups by playing an active role for aggregators [16].

This type of BM is essentially dependent on the special technology features of the virtual power plant (VPP) or the peer-to-peer (P2P) trading platform [1].

This virtuality requires a high level of coordination between the technology platform and the financial transactions that occur between utilities and customers; in addition, it requires control over the sale of electricity in wholesale energy markets [1,26]. As a result, in order to facilitate the transaction, a technology provider or a utility acts as a broker or a network provider.

The utility operates a number of small-scale generation units using virtual power plants for centralized management. It allows prosumers to trade excess energy with their neighbors and to protect themselves from retail market volatility [27,28]. It is possible to install the power plants on the rooftops of the customers' buildings or in the vicinity of the consumption sites [5,16].

### 2.2. Intermediaries

Intermediaries have an impact on new policies and market practices across the state, market, and civil society, and they assist the actors in developing sustainable practices and in adapting to new green technologies [29,30]. Based on the influential mechanisms of the intermediaries, the various types of intermediaries have been classified into niche development intermediaries and new modern intermediaries.

### 2.2.1. Niche Development Intermediaries

Intermediaries play an important role in the niche development process by ensuring the continuity and legitimacy of grassroots initiatives [29].

The role of niche development intermediaries can be divided into three categories: configuring, brokering, and facilitating [30].

Configuring is the process of assisting communities in adjusting to new BMs, interpreting usage, and adapting to new technologies.

Brokering is the networking of affairs in order to obtain support from various local stakeholders for the embedding of projects into local settings.

Finally, facilitating intermediaries facilitate activities such as ensuring information flows, knowledge sharing, and capacity building through financial or organizational and bureaucratic support [31].

### 2.2.2. New Modern Intermediaries

In the new CE development trends, the role of the intermediaries has shifted toward more activities, such as those of aggregators and third parties.

Aggregators are legal entities whose primary function is to optimize the production and consumption of electricity generated by prosumers. Because such intermediaries exist, market participants sell and buy electricity in a manner similar to that of a sharing economy, and there is no centralized oversight for local CE, such as microgrids [25,28].

Aggregators combine networks of distributed generation and provide brokerage services linking distributed CE and incumbent wholesale markets in the virtual power plant business models. In peer-to-peer business models, aggregators connect suppliers and buyers and enable trading among customers outside of the wholesale market [1].

Third-party financing is the preferred method for financing on-site renewable energy production. A third party is a company or an individual who hosts the renewable system, and the asset is owned by other investors, such as utilities, social entrepreneurs, and non-profit local governments [1].

Third-party or energy service companies benefit from a community's energy savings [16], which are typically based on different leasing contracts.

Leasing can be combined with any of the four different types of energy services. The first option is to grant licensed third-party access to network assets and power grid operations. The second type is a combination of network structure and energy supply infrastructures. The third type is to share collected data, analyze them, and increase network reliability. Finally, data and energy supplies are grouped for use in a market or application [32,33].

### 2.3. Policy Instruments

Some of the policy instruments and regulations in support of CE businesses have been introduced through fiscal incentives and payment-based instruments, such as FiTs, net metering, grid connectivity, low-interest loans, quota systems, and green certificates. FiTs are typically used to incentivize the adoption of renewable energy sources, such as solar or wind power, by compensating residential consumers and prosumers for any excess electricity generated and fed back into the grid. The payment is usually based on various tariffs, such as a feed-in premium above the market price or earnings from an aggregator or direct selling at the market price.

Net metering, on the other hand, incentivizes energy self-consumption by reducing electricity tariffs and providing tax exemptions to those who generate their own electricity using renewable sources. This allows consumers to offset their energy usage against the energy they generate, which results in lower electricity bills [13].

Pilot projects involve the process of scaling up CE and learning. The replicability of small and local energy community projects as pilot projects on a national and global level leads to some learning effects that increase the acceptance of CE projects and decrease the social indifference [16].

There are various sources of low-interest loans for renewable energy projects, including government programs, private lenders, and nonprofit organizations. The terms and

conditions of these loans may vary depending on the lender, but typically, they offer lower interest rates than traditional loans, longer repayment periods, and more flexible terms [34].

In terms of grid connectivity, the integration of renewable energy sources into the grid requires careful planning and coordination to ensure that the electrical system remains stable and reliable. To manage the variability, grid operators use a range of tools and technologies, including energy storage systems, demand response programs, and advanced forecasting techniques [35].

Regarding the green certificate, it is issued to track the renewable energy generation and to allow the owners of the certificate to claim the environmental benefits of that generation. These certificates can then be bought and sold on the open market, allowing companies and individuals to support renewable energy generation even if they are not directly using it themselves [36].

## 3. Research Design and Methodology of the Research

The methodology of this study is depicted in Figure 2, which also indicates the theoretical and practical phases of the research.

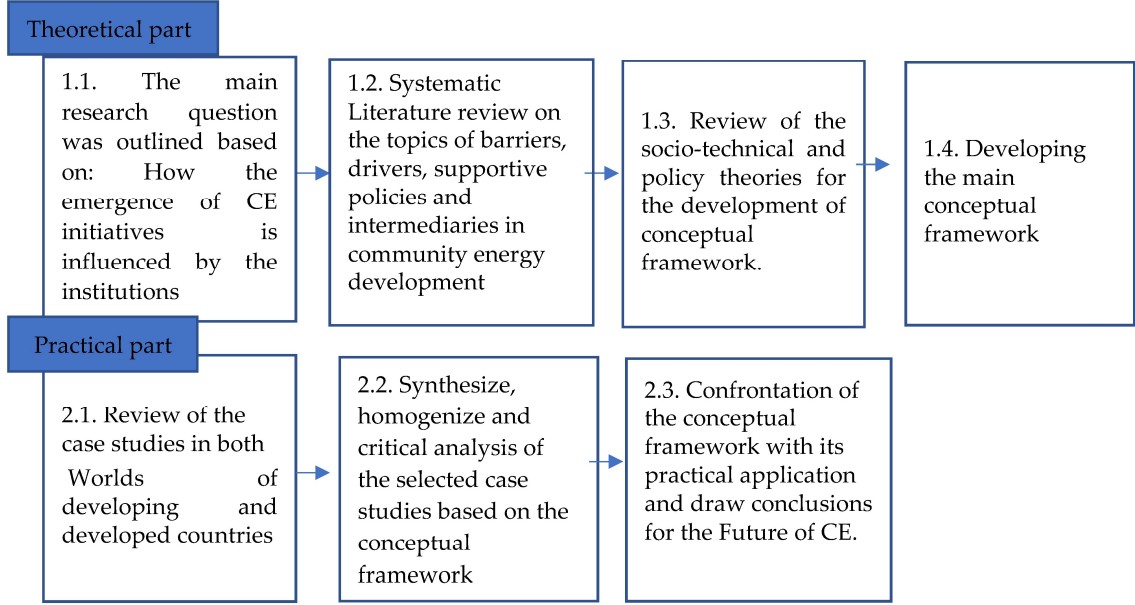

**Figure 2.** Research design and methodology of the research.

The theoretical phase of study: in the first step the following generic research question was outlined.

How is the emergence of CE initiatives influenced by the institutional environment?

Then, based on the main question, three additional sub-questions were considered to guide and frame the research.

(1) Which kinds of policy support instruments and intermediary partnerships account for the emergence of CE BMs?
(2) How and what types of BMs were developed for the selected cases?
(3) What is the mechanism that explains the sustainable energy communities in the selected cases?

In the second step, a systematic literature review, including various aspects of the conceptualization, BMs, drivers, and enablers of the energy communities, was conducted.

The author examined titles, keywords, and, in particular, abstracts to determine the relevance of the publications to the subject of business models for community energies. In order to avoid overlooking relevant studies, the author also examined the citations of the papers; consequently, a few additional studies were added.

In the third step, the author thoroughly went through all the articles to categorize them according to the research questions and to develop the conceptual framework (as described above, in Section 2).

In the fourth step, the collected information was synthesized and homogenized in accordance with the conceptual framework, and each case study was fully described in terms of the intermediaries and policy support, highlighting their core activities, the market and policy challenges they seek to address, and the competitive advantage they offer.

The practical phase of study: the case studies were selected based on their distinctive characteristics and examined in terms of their legal structure, geographic scope, ownership, activities, and actors. Their inclusion was intended to illustrate the real configurations, practical operationalization details, most common arrangements, and perceived implementation barriers.

In each of the case studies, an attempt was made to select the most successful projects in the selected countries. However, it is possible that different kinds of community BMs were selected in the same country, especially in the EU countries. The size of Europe's RE cooperative sector varies greatly. Although the RE cooperative model is well established in some countries, it is still a minor player in others.

The choice of case studies in the developed countries was based on four criteria: renewable energy units were applied in the project; advanced ICT was used; the project was carried out in Europe; and project information was available to the general public. Furthermore, the selection of European countries was based on their shared history of national policies supporting energy communities based on renewable energy deployment, according to the REN21 [15,37]. Then, four countries, Denmark, Germany, Belgium, and the UK, were selected because they have various primary support systems for RE development: feed-in tariffs have been implemented in Germany and Denmark, whereas trade-based quota systems are mostly used in the UK and Belgium [38].

In the selection of the developing country cases, the criterion was the differentiation of projects in terms of the business model and the stakeholders initiating REC in their neighborhoods. One similarity among the selected regions in the developing countries was the lack of access to electricity, which has been acknowledged as a positive experience by a number of international institutions and researchers [18].

The selected cases in the developing countries are located in rural Central America, South Africa, Iran, and Indonesia with analyses and detailed descriptions of their intricate structures.

## 4. Results

This section presents analyses of various cases in developing and developed countries using the conceptual framework.

### 4.1. Selected EU Countries

#### 4.1.1. Germany

In Germany, the introduction of the EEG in 2000 and the financial incentives such as FiTs and market premiums reduced the risk of individual investments in renewable energy development and facilitated collective ownership models [39,40]. The provision of low-interest loans with preferential conditions and priority grid access for CE initiatives were other supportive policies of CE initiatives [41].

Germany also has a tradition of cooperatives for local energy activism, such as bioenergy villages, which were launched by the government in 2005. Furthermore, a sufficient number of individuals are concerned about climate change and environmental protection.

One example is the anti-nuclear movement, which began in the early 1980s (following the Chernobyl disaster), when protest demonstrations prevented the construction of nuclear power plants [39,40].

Municipality Energy Community Projects

The energy cooperatives in Germany were mostly founded by small groups and local people, municipalities, and private companies involved in electricity supply and energy services. They have the potential for community storage as well as community-based concepts such as power-to-heat, virtual power plants, and microgrids [42]. More than two-thirds of the cooperative installations are on the roofs and estate properties of municipalities [43].

Several municipalities in Germany encouraged the large-scale direct participation of local citizens for the benefit of the community. In 2008, the University of Jena started supporting an energy cooperative initiative to buy back third-party shares in the municipal company. Citizens could purchase up to 10% of the shares of the municipal energy company under this plan. As a direct result of this plan, the first cooperative energy company, BürgerEnergie Jena, was founded and the extensive cooperation of local citizens, even with a minimum deposit of EUR 500, was promoted. The citizens not only benefited financially from owning a stake in the local utility, but they also had an influence on the commercial strategy of the community (Cooperative Energy Jena).

Another example is the Wolfhagen CE, where the local municipality supported the formation of a citizen cooperative with 14,000 residents. This cooperative owned 25% of the total capital of the municipality, and it had two representatives from the cooperative sitting on the nine-member supervisory board of the municipality. In 2003, the municipality director persuaded local politicians to accept a 20-year privilege contract for network distribution and to municipalize its electricity grid [43,44]. Table 1 provides a summary of the CE BMs in Germany.

**Table 1.** CE BMs in Germany, intermediaries and policy instruments.

| Country | | Germany |
|---|---|---|
| **Institutional environment** | **Policy support** | • FiTs<br>• Low-interest loans with preferential conditions<br>• Priority to the grid access |
| | **Business model 1** | **Centralized:**<br>Municipality energy community projects, BürgerEnergie, Jena |
| | **Intermediaries** | **Configuring:**<br>• Municipality,<br>• Jena university<br><br>- Citizens could buy up to 10% shares of municipal energy company<br>- Citizens have influence on the commercial strategy of the community |
| | **Business model 2** | **Centralized:**<br>Municipality energy community projects, Wolfhagen |
| | **Intermediaries** | **Configuring:**<br>• Municipality<br>• Citizen<br><br>- Citizens owned 25% of the municipality capital.<br>- There were two representatives of the cooperative sitting in the nine-member supervisory board of the municipality. |

### 4.1.2. Belgium

The electricity generation in Belgium is dominated by a former state monopoly and an incumbent company. Belgium plans to increase renewable energy generation to reach around 18% of total final consumption by 2023 [45].

The dominant policy support for renewable energy was the quota system based on the trade of certificates, but due to saturation of the green certificate market in 2011–2012, income for RE cooperatives decreased, and there is only a net-metering system as a remuneration scheme for surplus energy [46].

Belgium has a long history of cooperation but weak anti-nuclear movements. However, the opposition to nuclear waste in some areas led to the formation of cooperatives in that area.

The cooperative is mostly configured as a top-down cooperation through investor-owned companies [38].

### Joint Investment Cooperative: Coopem

The Coopem project was launched in collaboration with a group of citizens and two other partners. Thirty percent of the investment belonged to a green investment company, and 15% of the total share was owned by the municipality; the majority (55% of the stake) belonged to the citizens of Mouscron. The Coopem subsidized the initial costs of installments and regional solar payments, which were granted for a period of five years, to make the installation of solar PV panels more accessible and affordable. This community completely managed the technical and administrative processes, along with the purchase of equipment from local suppliers. Local businesses were also offered a leasing plan for the installation of PV panels. This plan finances 90 percent of the initial investment and is repaid over a period of ten years through the sale of green certificates [47].

### Smart Grid Development: Buurzame Stroom

Buurzame Stroom was initiated in cooperation with local partners in the city of Ghent. The target participants in this project were vulnerable, elderly, and migrant residents with different ownership structures. The consortium included two energy cooperatives, (Ecopower and EnerGent), Ghent University, a social protection association, and the distribution operator.

The plan of this consortium was to maximize local generation and benefits without any distribution to the electricity grid. The two cooperatives of Ecopower and EnerGent had different missions. Ecopower served as an aggregator by controlling demand and supply through the use of smart meters and open data in order to give homeowners more control over their energy consumption. The EnerGent collaboration provided incentives to local residents to invest in solar power panels in conjunction with the electric car sharing unit to store the excess power generated by electric vehicles and charging stations. Ghent University played a neutral role as a trusted partner, while the social protection institution served as an intermediary, reaching out to vulnerable households for profit sharing. The municipality played the role of coordinator and supporter between various partners [48]. Table 2 provides a summary of the CE BMs in Belgium.

**Table 2.** CE BMs in Belgium, intermediaries and policy instruments.

| Country | | Belgium |
|---|---|---|
| **Institutional environment** | **Policy support** | Quota system based on trade of certificates, Net metering |
| | **Business model 1** | **Distributed:**<br>Joint investment cooperative,<br>Coopem |
| | **Intermediaries** | **Third-party:**<br>• Green investment company<br>• Municipality<br><br>- Technical and administrative help for the panel instalment<br>- Financing 90% of the initial investments through the leasing plan which is paid back with selling of green certificates |
| | **Business model 2** | **Distributed:**<br>Smart grid development<br>Buurzame Stroom |
| | **Intermediaries** | **Aggregator and facilitating:**<br>• Two energy cooperatives, (Ecopower and EnerGent)<br>• Social protection association<br><br>- Two energy cooperatives serve as aggregators for smart metering and provide incentives for citizens to invest.<br>- Solar power panel installations for vulnerable, elderly, and immigrant residents<br><br>Social protection association for vulnerable household introduction |

4.1.3. UK

The UK was the earliest adopter of CE and communities owned 4% of solar PV installations in 2009; this was supported by the policy instruments of subsidization, the grant mechanisms of FiTs, a quota system, and tax relief [49]. However, since 2010 the FiT has only been applied to power plants smaller than 5 MW, and the tax relief schemes for communities have been unavailable since 2014 [15,50–52].

In the United Kingdom, volunteer experts and local governments as investors, as well as schools or community centers as host owners, collaborated on community buildings. Community buildings, as third-party premises, benefited from public subsidies in the form of FiTs and PPAs on distributed solar PVs for community buildings; these were partially offset by the selling of electricity to the owners of the premises [49].

Federation and Social Enterprises-Based Energy Communities

In the city of Plymouth, the founding members of a community benefit society known as "Plymouth Energy Community", equipped the rooftops of 32 schools and communities for the installation of solar PV projects. Municipal buildings, hospitals, and schools were the first priorities of the energy cooperatives due to their unchangeable conditions over a long period of time.

In order to tackle the energy poverty of vulnerable households, the Plymouth Energy Community established a home energy team that provided advice on switching energy suppliers and reducing consumption. After extensive public campaigns, the city provided a start-up loan for the new business plan of "A home energy team". As a direct result of implementing this plan, complete control was transferred to a board consisting of

volunteers from the local community, and the number of funding members increased from 100 to 1200 [43,53].

Energy4All is a federation of cooperatives in the UK which assists local groups in raising funds to establish cooperatives. This federation supported schools and communities in launching community projects. One example is the start-up of the Edinburgh community, which raised EUR 2 million for the installation of 24 solar PV arrays on public buildings in Edinburgh. This project was a collaboration between the municipality of Edinburgh and a public education campaign targeting children. The residents who purchase a share also receive a fixed return on their capital (5% per year) [54].

Following the changes to feed-in tariffs in England in 2013, this group founded the schools' energy cooperative and integrated renewable energy and energy efficiency projects. The solar photovoltaic panels on these forty-four schools were owned and operated by this cooperative. The profits and energy savings were returned to the schools, lowering their monthly bills. Energy4All assisted schools with both administrative tasks and the monitoring of their systems [55,56]. Table 3 provides a summary of the CE BMs in the UK.

**Table 3.** CE BMs in UK, intermediaries and policy instruments.

| Country | | UK |
|---|---|---|
| **Institutional environment** | **Policy support** | Subsidization, FiTs, quota system, and tax relief |
| | **Business model 1** | **Centralized:**<br>**social enterprise-based BMs**<br>Plymouth community |
| | **Intermediaries** | **Configuring and Brokering:**<br>• Local communities<br>• Plymouth energy community<br>- The Plymouth energy community empowers the communities to create an affordable renewable energy system in municipal buildings and hospitals and on school roofs. |
| | **Business model 2** | **Centralized:**<br>**Federation-based energy communities:**<br>Energy4all |
| | **Intermediaries** | **Facilitating and Configuring:**<br>• Federation of energy 4all<br>• Volunteer experts, local authorities, and councils<br>- The energy 4All supports local groups to raise money for increasing cooperatives.<br>- Residents who purchase a share also get a fixed return on their capital (5% per year).<br>- Profits and energy savings were paid back to schools, which reduced their monthly bills |

### 4.1.4. Denmark

Denmark aims to have all of its electricity and heat supply based on renewable energy by 2035; similarly, all of its transportation consumption will be based on renewable energy sources by 2050.

Denmark has historically been the pioneer of wind energy communities due to the cooperative nature of its wind energy provision [50]. In Denmark, community action began in the 1970s with a large share in onshore wind energy projects, and by 2002, over 150,000 households owned shares in wind power cooperatives [57].Typically, CE in Denmark is driven from the "bottom-up" and has influenced the political process to the extent that CE projects are mostly initiated by the collaboration of energy service entities rather than private companies [50,51].

The national funding of local energy, environmental offices, and local energy companies provided crucial procedural support for renewable energy initiatives and local communities [51].

There has been a long tradition of cooperation and strong, successful anti-nuclear movements in Denmark that has aided in the search for alternatives for the energy sector [38].

Jointly Owned Energy Infrastructure

The Middelgrunden Wind Cooperative is the first CE in Denmark. The Copenhagen Environment and Energy Office (CEEO) and a group of local people formed the wind cooperative model as a local electric utility. It was the largest offshore farm in the world in 2000, with 20 turbines. The municipal utility company now owns half of the wind farm, while the remaining 50% is owned by a thousand Middelgrunden Turbine Cooperative investors. The cooperative functions according to a democratic governance model, with each member having one vote regardless of the number of shares owned [58,59].

In Denmark, the community project on the Samsø islands is a great example of energy company integration in the process of project development. The three main focal points of the Samsø CE were SEEO (Samsø energy and environment office), SEC (Samsø energy company), and the PlanEnergi company. While the SEEO's main responsibility was the promotion of renewable energy by providing information and advice, the SEC's objectives were the implementation of wind turbines and district heating.

The PlanEnergi company had an important role as the intermediary between the external context and the local interests of the Samsø community. It informed the chairman of the local business network about the REI national master plan and translated national goals and guidelines into the local actions of CE [50,51]. Table 4 provides a summary of the CE BMs in Denmark.

### 4.2. Developing Countries

#### 4.2.1. Rural Central America

The three successful cases of Panama, Aprodelbo, and Coopeguanacaste were studied in the rural areas of Central America. A hybrid PV–wind–battery power plant in Panama provided electricity to the local primary school.

Regional and Voluntary-Based Communities

The installation of the power system was well coordinated by the Technological University of Panama (UTP) and the school parent association (SPA). The SPA and the schoolteachers had an impact on the decisions that were made in this community, and the maintenance capabilities were built with the assistance of UTP engineers. There was no legal ownership of the community members' power system assets [18].

Aprodelbo is the second case of an energy community in rural Central America, where a small micro-hydro turbine of 235 kw was installed by the association of a non-profit organization and an electricity provider for the small town of 3000 people. In this case, the ongoing operation was supported by a local NGO and periodic monitoring was conducted by external parties and the national electricity regulator. The technical design was the responsibility of local technicians and engineers. The community kept the power system, distribution network, and legal ownership of the asset [60].

The third case, Coopeguanacaste, is an electric cooperative that was founded by local leaders with financial support from the government's alliance progress program. Grid extensions, micro-hydro power plants, and PV solar home systems were the rural electrification solutions. For a period of 15 years, a subcontracted company using the build–lend–transfer model was responsible for the design, construction, and operation of the micro-hydro power plants. In this case, there was high bankability with local and financial institutions, and there was periodic monitoring by the national electricity regulator [61].

In all of the abovementioned cases, stable and long-lasting local and social structures aided in the implementation and operation of RE power plants and in both the technical design and the maintenance responsibilities.

**Table 4.** CE BMs in Denmark, intermediaries and policy instruments.

| Country | | Denmark |
|---|---|---|
| **Institutional environment** | **Policy support** | • Green certificates supersede the feed-in tariff in 2000 <br> • The national funding of local energy, environmental offices, and local energy companies provided crucial procedural support for the renewable energy initiatives and local communities |
| | **Business model 1** | **Decentralized:** <br> Jointly owned energy infrastructureSamsø islands |
| | **Intermediaries** | **Brokering, Configuring, Facilitating** <br> • SEEO (Samsø energy and environment office) <br> • SEC (Samsø energy company) <br> • The company of PlanEnergi <br><br> - SEEO's main task was to promote renewable energy <br> - SEC's objective was to implement specific projects, namely wind turbines and district heating. <br> - PlanEnergi company was an intermediary between the external context and the local community |
| | **Business model 2** | **Centralized:** <br> Jointly owned energy infrastructureMiddelgrunden Wind Cooperative |
| | **Intermediaries** | **Configuring:** <br> • Copenhagen Environment and Energy Office (CEEO) <br> • Group of local people <br> • Local utility <br><br> - The cooperative functions according to a democratic governance model, with each member having one vote regardless of the number of shared owned. <br> - The local utility provided technical and legal expertise. |

The political context around each case study was not supportive of community deployment. In rural Panama, for example, the national grid was extended to a limited area and very small stand-alone systems were not encouraged. On the other hand, in Nicaragua, there has been a long history of community activism for the handover of CE projects and private distribution companies. Furthermore, the high level of dependence on external people or organizations during the design phase did not allow for the improvement of local capabilities and subsequently the technicians were trained from the outside. In the case of Panama, for example, UTP engineers led technical designs, without any participation from community members, whereas in Costa Rica, a private company is in charge of the micro-grid operation. However, in Coopeguanacaste, the required skills are acquired internally by the community's residents [18]. Table 5 provides a summary of the CE BMs in rural Central America.

**Table 5.** CE BMs in rural Central America, intermediaries and policy instruments.

| Country | | Rural Central America |
|---|---|---|
| **Institutional environment** | **Policy support** | The political context around each case study was not supportive of the CRE community deployment, except for some financial support. |
| | **Business model 1** | **Decentralized:** Regional and volunteer-based communities:Panama |
| | **Intermediaries** | **Brokering, Configuring, Facilitating:** <ul><li>UTP (Technological University of Panama) engineer</li><li>SPA (school parent association)</li></ul> - The UTP engineers led technical design and maintenance. <br> - SPA and schoolteachers made central decisions. There was no legal ownership for community members. |
| | **Business model 2** | **Decentralized:** Regional and volunteer-based communities:Aprodelbo |
| | **Intermediaries** | **Brokering, Configuring, Facilitating:** <ul><li>Local NGO</li><li>Local technicians</li></ul> - The local NGO supported the operation <br> - Local technicians and engineers were responsible for the technical design |
| | **Business model 2** | **Decentralized:** Regional and volunteer-based communities:Coopeguanacaste |
| | | **Brokering, Configuring, Facilitating** <ul><li>local leaders</li><li>Supportive financial institution</li><li>The build–lend–transfer contracted company</li></ul> - Local leaders created electric cooperatives <br> - Financial accountability and managerial capabilities built in alliance with financial institutions. <br> - A company was subcontracted for the design, construction, and operation of the micro-hydro plants |

### 4.2.2. Iran

In Iran, a systematic plan for clean energy production from renewable sources has been in place since the beginning of 2010, and a FiT law for motivating private sector investment in renewable energy has been enacted [20]. Iran has a long tradition of engaging in non-profit activities and social cooperative behaviors in support of deprived people.

Social Enterprise Community

In Iran, around three thousand small-scale solar power plants have been built under the title "Blessing of the Sun" for low-income groups. According to a national plan that was initiated by the ministry of energy (MOE), the intermediary, the Imam Khomeini Relief Foundation, had the responsibility of establishing and operating these power plants for the underprivileged people who fell under the coverage of the Foundation in 18 provinces across the country. Microfinance services were utilized in order to fulfill this project's need for a source of funding. The people were financially disadvantaged and did not have access to financial resources; therefore, a banking service offered microfinance services. These services included the provision of small loans without the requirement of collateral to people who were interested in investing in solar PV. In accordance with the regulation

governing FiTs, the households in rural areas, nomadic areas, and low-income areas benefited from FiTs for the distribution of renewable electricity to the main grid [62]. Table 6 provides a summary of the CE BMs in Iran.

**Table 6.** CE BMs in IRAN, intermediaries and policy instruments.

| Country | | Iran |
|---|---|---|
| **Institutional environment** | **Policy support** | Long-term contracts of FiTs<br>National plan was initiated by ministry of energy |
| | **Business model** | **Centralized:** Social enterprise community |
| | **Intermediaries** | **Brokering, Configuring, Facilitating:**<br><br>• Relief Foundation<br>• Ministry of Energy (MOE)<br><br>The Relief Foundation intermediary has the responsibility of establishing and operating solar power plants with the help of contracted firms. |

### 4.2.3. South African Countries

The South African government began investigating renewable energy feed-in tariffs (FiTs) in 2009, but this plan was rejected in favor of competitive tenders.

In 2011, the South African government launched a competitive public program generally known as the Renewable Energy Independent Power Producer Procurement Programme (REIPPPP).

This plan channeled private sector expertise toward the development of renewable energy facilities and grid-connected power plants in South Africa [63,64].

Since the inception of this public–private partnership, more than USD 16 billion in private sector investment has been provided for 79 awarded projects, for a total of 5243 MW renewable energy power plants. Several bid winners and a rolling bidding process that incentivizes more participants were among the key design elements of this public–private partnership. To reduce the challenges associated with underbidding, all the bids were required to be fully underwritten by debt. The investors were assured of the benefits through the use of tariff caps. Complementary policies were also implemented in order to reduce administrative barriers and fast-track the program [65].

This program was highly beneficial for South Africa. Wind and solar power plant tariffs have fallen even lower than the price of new coal power plants. The economic objectives of this plan have resulted in the creation of new jobs, preferential procurement objectives, and foreign direct investment [66,67].

There were also some examples of energy community projects that did not result in sustainable energy generation. One example is the community projects in the western region of Cameroon. Private and social actors work together, along with the critical engagement of the government and NGOs. The NGOs promoted renewable energy technology and provided technical assistance for the conceptualization of the hydroelectric project. They also assisted in securing 70% of the funding from development partners. However, the inability of the villages to formulate socially oriented projects reduced their access to the funds provided by developers [68]. Table 7 provides a summary of the CE BMs in South African countries.

**Table 7.** CE BMs in South African countries, intermediaries and policy instruments.

| Country | | South African Countries |
|---|---|---|
| **Institutional environment** | **Policy support** | In 2011, introduction of the competitive public program of Renewable Energy Independent Power Producer Procurement Programme (REIPPPP) |
| | **Business model** | **Centralized:** Public–private partnership |
| | **Intermediaries** | **Configuring:** Private sector investors Public government 1. Private sector invests in renewable facilities in grid-connected power plants. 2. Process of bid winners and rolling bidding The investors were ensured about the benefits through connection with tariff caps. |

### 4.2.4. Indonesia

Indonesia is an example of a developing Asian country with rising industrialization and energy demands [69].

The three main actors in the country's electricity production are the national electric company (PLN), the independent power producers that generate electricity from renewable sources and feed it into the grid at the rate of national feed-in tariffs, and, finally, the rural cooperatives that have permission to generate and distribute electricity [70].

Donor-Based Energy Communities

In Indonesia, despite the challenges and failed attempts, there are very positive examples of community-based RE mini-grids based on mini-hydro power plants.

The cooperative model has a long tradition. In Indonesia, women were active cooperative members in the cases of both Cinta Mekar and Kamanggih.

These community energy projects were mainly led by private and non-profit actors such as IBEKA (the acronym stands for People Centered Economic and Business Institute), which was primarily responsible for empowering and training rural communities. Some international initiatives, such as Energizing Development (EnDev) or RewiRE, provided financial assistance to these CE projects as well [70].

The need to find private investors and excessive dependency on international donor agencies were identified as important economic drivers. Although most donor funds were allocated to the construction phase, other stages of training or operation faced challenges.

The economic benefits of selling electricity include the generation of sufficient savings to cover current costs and side expenditures, such as that for fixing broken equipment. Soft loans to create new productive activities were also required as a catalyst for the long-term success of cooperatives because income-generating businesses guaranteed the ability of the participants to pay the electricity fees and ensured the financial sustainability of the project.

Some examples of the business initiatives that were supported by the cooperatives were the production of medical bags for export or the banana flour production by the local women.

In both cases, electricity generation was a motivator for socio-economic improvement. The ownership of a community-based energy project gave a sense of empowerment to the participants, as did the spillover of such projects to other villages [69]. Table 8 provides a summary of the CE BMs in Indonesia.

**Table 8.** CE BMs in Indonesia, intermediaries and policy instruments.

| Country | | Indonesia |
|---|---|---|
| **Institutional environment** | **Policy support** | There is no special plan for developing renewable energy communities. |
| | **Business model** | **Decentralized:**<br>**Donor-based energy communities**<br>Cinta Mekar and Kamanggih in Indonesia. |
| | **Intermediaries** | **Brokering, Configuring, Facilitating:**<br>• International donor initiatives: Energizing Development (EnDev) or RewiRE<br>• Private and non-profit organization: IBEKA<br><br>- IBEKA empowered and trained rural communities<br>- International donors helped with the financial assistance. |

## 5. Discussion

This section is divided into two sections related to the CE BMs in developing and developed countries. First, a summarization table focuses on the current challenges and future extensions according to the conceptual framework; then, an extensive explanation is presented (Table 9).

**Table 9.** Current challenges and possible solutions and practices.

| Policy support focus | Industrialized Countries | Developing Countries | Focus on Current Challenges | Possible Solutions and Practices |
|---|---|---|---|---|
| | ● | ● | Inflexible market structures and uncertain feed-in tariff levels | 1. Flexible transition frameworks must be created to allow the existing projects to continue to operate.<br>2. Brokering and managing intermediaries can perform an enabler role, combining legal, commercial, financial and technical expertise through framework agreements to de-risk contractually complex PPAs, and to create replicable and financeable community business models.<br>3. Facilitating intermediaries assisted in carrying out feasibility studies and providing business services and policy advice to communities, while also interacting with policymakers to ensure the motivational side of regulations for energy projects |
| | ● | | Most of the existing projects date back several years, which means they do not comply with the rules formulated by the current European directives. | |
| | | ● | The lack of institutions (public and private) and authorities dedicated to supporting and encouraging the development of the new market structures which are necessary for the greater prosumer integration. | |
| | | ● | Centralized and top-down approach to energy system | |
| | | ● | Weak democracy and low citizen participation | |
| | ● | ● | High initial investment requirements and lack of financial resources during the project lifetime and long payback periods | |

**Table 9.** *Cont.*

| | | | |
|---|---|---|---|
| Business model focus | | • | End-users are rarely aware of the scope and benefits offered by participating in energy projects | 1. Configuring intermediaries are assisting communities in adjusting to new BMs, interpreting usage, and adapting to new technologies<br>2. Local energy systems are likely to change with the introduction of plug-in electric, hybrid, and vehicle-to grid-technologies. Rising penetration of electric vehicles will yield higher load as well as storage capacity for ICESs.<br>3. Advancing cheaper smart meter technology, the development of BMs associated with IoT and online platforms, providing flexibility to deal with the intermittency and fluctuations of variable renewable energy, and guaranteeing a reliable supply of electricity |
| | • | • | Cognitive barriers of utilities with existing business models, such as risk aversion of responsible individuals and lack of recognition of business opportunities. | |
| | • | • | Delays in the development of potential BMs due to the lack of information, knowledge, and training on the characteristics of the prosumers, P2P, smart grid models, energy markets and less availability of the physical and technological infrastructure. | |

### 5.1. Current Conditions and Future Extension in Industrialized Countries

Northwestern European countries are pioneers in the field of community-driven energy projects due to their national policies that encourage the development of decentralized renewable energy projects [71].

The ambitious target of renewable energy consumption resulted in political support for CE as well as financial incentives in terms of FiTs and market premiums, loan capital with preferential conditions, and priority access to the grid.

These economic supports provided greater financial security than other investment markets, which are subject to certain risks. They encourage citizens to install renewable energy production units by facilitating their grid connection, which encourages more people to install renewable energy production units. Furthermore, the high sensitivity of citizens to changes in the environment, a long-standing tradition of local energy activism, and the agendas of political groups were some of the other factors that contributed to the proliferation of CE [40].

Future smart neighborhoods will be able to optimize energy access in a variety of ways by sharing resources in a controlled manner. New trends in distributed cross-commodity energy management, as well as ICT perspectives such as the internet of things, artificial intelligence, and blockchain play important roles in sector-integrated energy systems. Networked prosumers can negotiate and create smart contracts that match local demand, supply, and storage. The timely management of distributed cross-commodity resources will be critical to the energy efficiency and stability of the smart city [72]. The primary objective of these models is to convert residential buildings into high-energy-efficiency community buildings by means of the management of energy consumption, accomplished through the application of technological advances in artificial intelligence. In order to accomplish this objective, the existing energy supply is being upgraded to new, complex photovoltaic, solar thermal, and cogeneration systems. This model has recently been implemented in Germany in the smart district of Karlsruhe-Durlach, and as a result, both primary energy consumption and CO2 emission levels were cut in half [42].

However, the elimination of some of these regulations led to a reduction in the number of people adopting the CE idea. Because of the withdrawal of FiTs (in the UK) or the expiration of a 20-year agreement (in Germany), clean energy projects are now vulnerable to the development of high-capacity RE power plants [73].

One possible solution is community-owned energy storage, which can restore the economic feasibility of CRE projects while also allowing low-income households to benefit from lower prices in comparison to those of energy provided by the national grid. Such

a business model necessitates additional technical and business expertise in the form of solar-plus-storage communities that collaborate with aggregators and local suppliers [49].

In developing countries, unfulfilled basic needs, a growing demand for energy, and the desire to bridge the gap between urban and rural areas are the main drivers of community BMs [12].

*5.2. Current Conditions and Possible Solutions in Developing Countries*

One of the most significant barriers to CE deployment in developing countries is the centralized energy system, which prohibits the deployment of CE in order to gain more authority in energy accessibility [74]. Most developing countries have top-down energy policies, and only a few have mandates at the city or municipal level. This energy structure is insufficient for participatory energy planning [75].

In sub-Saharan Africa, the dominant centralized energy system typically opposes democratized energy distribution. Communities are not sufficiently involved in microgrid project initiation, and the majority of projects are wholly owned by the government or elite organizations.

In many African and South Asian countries, rural electrification is typically based on mini-grid development and stand-alone systems. However, the power supply is frequently unreliable in these types of systems. Although some international institutions or donors have financed some pilot renewable projects for the grid connection in a limited period of time, it appears that a decentralized approach based on off-grid systems, such as solar PV, mini-hydro, or other renewable energy sources, is often the best option for providing energy services in such regions. Indeed, CE and local NGOs are an appropriate approach in response to local challenges for the operation, investment, and maintenance of energy systems [31,69].

The sophisticated BMs, such as peer-to-peer and smart grid models, which are applicable in industrialized countries cannot be replicated in developing countries, and the CE sectors have mainly faced challenges such as a lack of technical knowledge, financial barriers, and inadequate capacity building mechanisms. In these countries, unsupportive policies, a lack of appropriate regulatory frameworks, and a lack of institutional supports, such as policy advice or business services to develop and manage energy projects, hinder the advancement of novel CE BMs [76,77].

Regarding the abovementioned challenges, it seems that the presence of intermediaries such as developers that work 'in-between' other actors, such as energy providers, users, or regulators, is essential for the initiation of CE projects in developing countries. Various actors, including public authorities, NGOs, academic institutions, and networks, can serve as intermediaries to fill existing gaps and connect external drivers to the community [17,69].

In the cases of Panama and Aprodelbo, configuring intermediaries emerged in the form of university engineers who led the technical design and a subcontracted company that implemented a new BM of build–lend–transfer for the operation of micro-hydro plants.

In Iran's case, the role of brokering intermediaries facilitated co-design strategies for connecting energy groups with microfinance and vulnerable local communities.

In the case of Indonesia, the role of facilitating intermediaries assisted in the carrying out of feasibility studies and the providing of business services and policy advice to communities while also interacting with policymakers to ensure the motivational side of the regulations for energy projects [75]. In the case of Indonesia, a strong and active mediatory organization plays an important role in helping community projects thrive.

On the other hand, the engagement of an intermediary is needed in order to facilitate project success through "aggregation and learning", "establishing an institutional infrastructure", "framing and coordinating local project activities", and "brokering and coordinating partnerships with [outside] actors" [52,69].

## 6. Conclusions

As highlighted in the introduction section, this study focuses on the neglected factors of the institutional environment in terms of policy support and intermediaries for the CE business configuration. This paper presented an exploratory structure and examined some different cases in industrialized countries of the EU and in developing countries in South Africa, South America, and Asia to answer the questions of how regulatory frameworks and policy instruments could be considered to drive community participants and form community BMs under different institutional and contextual factors and how the role of the intermediary sheds light on opportunities for emerging CE BMs.

This examination of CE development might help policymakers and practitioners in better understanding the need for changes and in implementing the actions required to effect institutional changes.

The contribution of the research is to assist policymakers in understanding what the different forms of BMs are, how they can be configured, and what the potential influences of national policy support and the institutional environment on sustainable CE business are. This study helps decision makers to create an institutional setting that supports a low-carbon future by supporting them in understanding the dynamics of various CECs, how they have changed over time, and how they will affect current and future energy markets and systems. Policymakers should create clear and reliable legal framework conditions to create stable planning reliability that encourages new ventures. Current trends toward energy self-sufficiency, district heating with renewables, and community ownership present highly appealing research and business opportunities in developed countries.

In the northwestern European countries, the purpose of energy communities is to facilitate the production and consumption of energy in the most efficient manner. In this regard, actors are enabled through different BMs and effective technical and market instruments to provide a platform for the use of various types of smart technology and artificial intelligence.

In developing countries, which typically have centralized electricity systems and unreliable access to electricity in rural areas, the role of intermediaries is more prevalent in connecting consumers, entities, and public institutions. Overall, developing countries require contextualized CE strategies based on grassroots social movements that are driven by localized approaches and strong leadership. Overall, community energy projects necessitate a more accountable and democratic political structure that gives community members ownership and participation in policy and decision making.

In this regard, in developing countries it is necessary to transform the centralized energy systems into more decentralized ones, as well as to transform the current institutions and policies. In particular, a supportive legal framework in developing countries should be based on proven and tested laws in developed countries, but it should be appropriately tailored to differing specific situations and regions.

Furthermore, the smart energy systems and innovative CE BMs that been developed and proven in developed countries appear to be viable options for developing countries. These technological advances may alter the role of individuals in energy systems as they require higher levels of awareness, passion, and participation in society.

This research also contributes by offering examples of both traditional and modern forms of CE BMs in order to inform the communities and households seeking cleaner and more efficient energy solutions about their options and their potential to meet their future energy needs. This will encourage customer participation in more distributed and decentralized energy systems.

**Funding:** This research was funded by the German Research Foundation and the Open Access Publication Fund of TU Berlin.

**Institutional Review Board Statement:** Not applicable.

**Informed Consent Statement:** Not applicable.

**Data Availability Statement:** The data presented in this study are available on request from the corresponding author.

**Acknowledgments:** The author expresses gratitude to the referees for their insightful remarks and suggestions. The author also acknowledges Yusef Nasehzadeh and Zahra Tabatabayi for their guidance.

**Conflicts of Interest:** The author declares no conflict of interest.

## Abbreviations

| | |
|---|---|
| CE BMs | Community energy business models |
| FiTs | Feed-in tariffs |
| CE | Community energy |
| BMs | Business models |

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
