# Peer review of "Investigation of Community Energy Business Models from an Institutional Perspective: Intermediaries and Policy Instruments in Selected Cases of Developing and Developed Countries"

_sustainability, doi:10.3390/su15108423_

Round 1

Reviewer 1 Report

The study develops a conceptual framework comprising policy instruments and intermediaries in configuring different community energy business models and examines this framework in both industrialised and developing countries. The topic is interesting, but the paper is very difficult to read and understand its content. Accordingly, the following major updates are required:

1) Improve the English style;

2)  Improve the quality of figures and tables; For instance, figure 2 is not readable at all;

3) A significant update of the text, in terms of both formatting and typos fixing is required;

4) The author should emphasise the objectives of the research, as well as the difference with existing publications;

5) The author should explain in a more detailed way the conceptual framework in Section 3;

6)  Section 6 is very confusing. The author treats the subject in a non-schematic way and, consequently, it is very difficult to follow the thread of the discussion;

7) A table highlighting the major differences between industrialised and developing countries should be added

Author Response

Many thanks for your valuable review and comments.

1) Improve the English style;

I have edited the English style of the paper according to your suggestions.

2) Improve the quality of figures and tables; For instance, figure 2 is not readable at all.

I have also changed Figure 2 to make it more readable. and 

3) A significant update of the text, in terms of both formatting and fixing typos, is required.

The grammatical problems have been checked and corrected.

4) The author should emphasize the objectives of the research as well as the differences with existing publications.

The parts of previous research gaps and the objectives of this research are added to the paper for more clarification of the research necessity. Part 1.2 extensively explains the differences of this research with previous research and the novelty of this study.

5) The author should explain in a more detailed way the conceptual framework in Section 3.

I have considered this comment and extended the explanation of the conceptual framework.

Section 6 is very confusing. The author treats the subject in a non-schematic way, and consequently, it is very difficult to follow the thread of the discussion.

According to your feedback, more explanations of the results of the case studies with classification have been discussed.

7) A table highlighting the major differences between industrialized and developing countries should be added.

A table of challenges and applicable solutions in developing and industrialized countries has been added.

Reviewer 2 Report

The manuscript entitled “Investigation of Community Energy Business Models from An Institutional Perspective: Intermediaries and Policy Instruments in Selected Cases of Developing and Developed Countrie” creates and evaluates a conceptual framework comprised of policy instruments and intermediaries in designing diverse community energy business models in both the worlds of north-western European countries and selected cases of developing countries. The manuscript has the following issues to be resolved:

·         Authors should include underdeveloped countries in the study.

·         In figure 2 some contents are not visible due to the overlapping of the figure.

·         Manuscript has grammatical errors and incomplete words even in the title of the manuscript “Countrie”.

·         Authors should discuss the role of government in policy implementation to popularize to energy projects in developed and developing countries.

Author Response

Many thanks for your comments and feedback.

  • In figure 2, some contents are not visible due to the overlapping of the figure.
    • I have changed Figure 2 to be more readable.
  • The manuscript has grammatical errors and incomplete words, even in the title “Countrie”.
  •                      I have checked and corrected the grammatical mistakes and English style.
  • Authors should discuss the role of government in policy implementation to popularize to energy projects in developed and developing countries 
    • According to your suggestion, I have added two paragraphs at the conclusion, for more clarification of the role of governments and policymakers in CE BMs development.

Reviewer 3 Report

1. There are many grammatical, sentence structure, and spelling errors all over the manuscript. Please proofread the manuscript properly.

2. Not clear what is the aims and goals of this article. The thought process is unclear, and the research novelty of the work is unknown since many of the results are already known by mainstream media.

3. Literature review does not clearly state the research gaps, nor is it done in a systematical approach whereby the 3 subsections do not link up. Readers will be confused on what are the existing business models, the impacts of those models, etc.

4. The core research elements only take up 1 page. It is not only high level and unclear, but readers also do not understand the novelty of the research since there are many existing works, as cited in sections 3 and 4.

5. Why are these countries selected in the case study? What are the steps done to validate the findings? Since most of the results are cited from existing work, what is the scientific contribution of this work?

Author Response

1. There are many grammatical, sentence structure, and spelling errors all over the manuscript. Please proofread the manuscript properly.

According to your feedback, I have completely edited the manuscript in English.

2. It's not clear what the aims and goals of this article are. The thought process is unclear, and the novelty of the research is unknown since many of the results are already known by mainstream media.

I have extensively added more explanation, regarding the research  gaps, contribution, and novelty of this research, in Section 1.2.

In all of the aforementioned research studies, scholars have debated the barriers, enablers and the relations between business models and policy support. While previous researchers derive their valuable findings from empirical analyses of existing EC projects, but none of them investigates or focuses on conceptualizing CE structures from an institutional perspective and an in-depth debate about the relationship between institutional arrangements (policies, process, intermediaries) and community energy business models has been absent.

This work will fill this gap. This research provides insight on how the emergence of CE initiatives is influenced by policy support instruments and the role of the intermediary between communities in the social, private, and public sectors [19,20].The main objective of this study is to investigate the mechanisms through which the actions of intermediaries (both traditional and new) in relation to policy instruments (Fits, net-metering, pilot projects) interact iteratively to enable the emergence of new CE practices.

Hence, this study has developed a conceptual framework of community energy business models from an institutional perspective, which will help to better understand the mechanisms through which intermediaries and policy instruments interact to enable the emergence of new CE practices.

This research will answer the following research questions:

Which kinds of policy support instruments and intermediary partnerships account for the emergence of CE BMs? 2) How and what types of BMs were developed for the selected cases and? 3) What is the mechanism that explains sustainable energy communities in selected cases?

For a more accurate comparison of contextual factors, selected case studies from both developing and developed nations are chosen. The contribution of this research is to employ an empirical method to compare developing and developed countries in northern European countries and the aim is to explain the differences in institutional arrangements between these two groups of countries.

The novelty of this work is the interdisciplinary approach to CE structures.

The subject areas of policy sciences, business studies, institutional frameworks, renewable energy sources, and the latest trends in technology were all brought together. The cognition of how these areas affect each other allows for a comprehensive understanding of the EC concept, its complexity, and the possible direction of its evolution.

Furthermore, the paper juxtaposes theory and practice, first describing the conceptual institutional framework of community energy, business models, and the role of intermediaries. Then presents the practical application of EC projects and their real market designs. This study brings together researches from both developing and developed countries as well as a conceptual framework for the business models that have been used to deploy community-based energy projects."

3. The literature review does not clearly state the research gaps, nor is it done in a systematical way whereby the three subsections do not link up. Readers will be confused about the existing business models, the impacts of those models, etc.

According to your comment, I have made some changes to the literature review part and extended the explanation of the conceptual framework in Section 2. Now, the literature review is in accordance with the conceptual framework explanation.

4. The core research elements only take up 1 page. It is not only high level and unclear, but readers also do not understand the novelty of the research since there are many existing works, as cited in sections 3 and 4.

The novelty of the research is completely discussed in section 1.2 and the research method and methodology are explainded in section 3 with more extended explanation.

The novelty of this work is the interdisciplinary approach to CE structures.

The subject areas of policy sciences, business studies, institutional frameworks, renewable energy sources, and the latest trends in technology were all brought together. The cognition of how these areas affect each other allows for a comprehensive understanding of the EC concept, its complexity, and the possible direction of its evolution.

Furthermore, the paper juxtaposes theory and practice, first describing the conceptual institutional framework of community energy, business models, and the role of intermediaries. Then presents the practical application of EC projects and their real market designs. This study brings together researches from both developing and developed countries as well as a conceptual framework for the business models that have been used to deploy community-based energy projects.

5. Why are these countries selected in the case study? What are the steps done to validate the findings? Since most of the results are cited from existing work, what is the scientific contribution of this work?

the process of case study selection  is described completely in the part of research method (section 3).

The methodology of this study is depicted in Fig.2, which also indicates the theoretical and practical phases of the research.

The theoretical phase of study: at the first step, this generic research question was outlined:

How the emergence of CE initiatives is influenced by the institutional environment?

Then, based on the main question, three additional sub-questions were considered to guide and frame the research. Which kinds of policy support instruments and intermediary partnerships account for the emergence of CE BMs? 2) How and what types of BMs were developed for the selected cases and? 3) What is the mechanism that explains sustainable energy communities in selected cases?

In the second step, a systematic literature review, including various aspects of conceptualization, BMs, drivers, and enablers on the energy communities was conducted.

The author examined titles, keywords, and, in particular, abstracts to determine the relevance of publications to the subject of business models for community energies. In order to avoid overlooking relevant studies, the author also examined the citations of papers discovered and consequently, a few additional studies were added.

In the third step, the author thoroughly went through the entire articles to categorize them according to research questions and develop the conceptual framework (abovementioned in section 2).

In the fourth step, the information collected was synthesized and homogenized in accordance with the conceptual framework and each case study was fully described in terms of intermediaries and policy support, highlighting their core activities, the market and policy challenges they seek to address, and the competitive advantage they offer.

The practical phase of study: the case studies were selected based on their distinctive characteristics and examined in terms of their legal structure, geographic scope, ownership, activities and actors. Their inclusion was intended to illustrate real configurations, practical operationalization details, the most common arrangements and perceived implementation barriers.

In each of the case studies, it is attempted to select the most successful projects in the selected countries. However, it is possible that different kinds of community BMs were selected in the same country, especially in EU countries. The size of Europe’s RE cooperative sector varies greatly. Although, the RE cooperative model is well-established in some countries, it is still a minor player in others.

 The choice of case studies in developed countries was based on four criteria: renewable energy units are applied, advanced ICT is used, the project is carried out in Europe and project information is available to the general public. Furthermore, the selection of European countries was based on their shared history of national policies supporting energy communities based on renewable energy deployment, specifically according to the REN21[15,37]. Then,  four countries of Denmark, Germany, Belgium, and the UK were selected, because they have various primary support systems for RE development: Feed-in tariffs have been implemented in Germany and Denmark, whereas trade-based quota systems are mostly used in the UK and Belgium [38].

In the selection of developing country cases, the criterion was the differentiation of projects in terms of business model and stakeholders initiating REC in their neighbourhood. One similarity among selected regions in developing countries was the lack of access to electricity, which has been acknowledged as a positive experience by a number of international institutions and researchers[18].

Reviewer 4 Report

The methodology used to summarize drivers and barriers by country is not clear. So it cannot be excluded that the findings are not objective or, at best, limited.

In addition, I recommend to revise the use of tables and figures to ensure all text and information is clear. See Fig. 2, for example-this should not be seen in a published article.

Author Response

Many thanks for your comments and feedback.

  1. The methodology used to summarize drivers and barriers by country is not clear. So it cannot be excluded that the findings are not objective or, at best, limited. The methodology and the conceptual framework are completely explained in Sections 2 and 3. In addition, the novelty of this research, which differentiated it from others, is explained.

Case study selection is described completely in the section on research methods and methodology (Section 3).

The practical phase of study: the case studies were selected based on their distinctive characteristics and examined in terms of their legal structure, geographic scope, ownership, activities and actors. Their inclusion was intended to illustrate real configurations, practical operationalization details, the most common arrangements and perceived implementation barriers.

In each of the case studies, it is attempted to select the most successful projects in the selected countries. However, it is possible that different kinds of community BMs were selected in the same country, especially in EU countries. The size of Europe’s RE cooperative sector varies greatly. Although, the RE cooperative model is well-established in some countries, it is still a minor player in others.

 The choice of case studies in developed countries was based on four criteria: renewable energy units are applied, advanced ICT is used, the project is carried out in Europe and project information is available to the general public. Furthermore, the selection of European countries was based on their shared history of national policies supporting energy communities based on renewable energy deployment, specifically according to the REN21[15,37]. Then,  four countries of Denmark, Germany, Belgium, and the UK were selected, because they have various primary support systems for RE development: Feed-in tariffs have been implemented in Germany and Denmark, whereas trade-based quota systems are mostly used in the UK and Belgium [38].

In the selection of developing country cases, the criterion was the differentiation of projects in terms of business model and stakeholders initiating REC in their neighbourhood. One similarity among selected regions in developing countries was the lack of access to electricity, which has been acknowledged as a positive experience by a number of international institutions and researchers[18].

Round 2

Reviewer 2 Report

The authors submitted a revised manuscript, "Investigation of Community Energy Business Models from an Institutional Perspective: Intermediaries and Policy Instruments in Selected Cases of Developing and Developed Countries." . The authors' responses to all the comments were satisfactory. Now paper considers the pulication.

Author Response

Thank you for your confirmation.

Reviewer 3 Report

1. Thank you for spending the time to reply to each question and to amend the manuscript. Some minor paragraph and alignment issues exist. Otherwise, it is good to go.

Author Response

Thank you for your confirmation.

I also went over the manuscript again and corrected various typos and figures.

Reviewer 4 Report

-Please spell check the whole paper. For instance, Indonesia is written as Indonisia.

-The abstract: spell out the country assessed exactly, not just which geographical area

-The introduction, first statement: please revise, I do not understand its meaning or relevance or how that would be the opening statement. 

-CE abbreviation might be ambiguous in this field. IT is for instance usually used to denote choice experiments as part of stated preferences studies. Suggest to change to CEn or similar.

-Figure 1: Can the quality be improved? It can be made neater and of better quality resolution. More importantly, is the study putting forward this conceptual framework? How much is already present in the literature? This should be clearly outlined in the text and in the Figure as well.

-Figure 2: Do we really need it? Can't we just discuss in the main text? This looks like a rather standard research design. In the text, provide a stronger justification for the countries object of assessment.

-Tables 1-8: add one column where you list the references in support of each claim made. 

-In the conclusions, you use the verb "surveyed". Was a survey conducted? No, so please amend.

-In the conclusions, can we have stronger recommendations, or is not possible based on the research conducted? 

For instance: A supportive legal framework in developing countries should be based on proven and tested laws in developed countries,  but should be appropriately tailored to differing specific situations and regions.

How is this adding value to previous research? 

Author Response

Many thanks for your precise review and comments.

  1. Please spell check the whole paper. For instance, Indonesia is written as Indonisia.
  • It is done.
  1. -The abstract: spell out the country assessed exactly, not just which geographical area
  • I have added exactly the names of the countries at the abstract.
  1. -The introduction, first statement: please revise, I do not understand its meaning or relevance or how that would be the opening statement. 
  • -This section was updated with more clarity and a better explanation.

3. -Figure 1: Can the quality be improved? It can be made neater and of better-quality resolution. More importantly, is the study putting forward this conceptual framework? How much is already present in the literature? This should be clearly outlined in the text and in the Figure as well.

  • I altered the image and tried to produce a better figure with good quality.
  • I highlighted “the proposed conceptual framework produced by the author” in the text and the figure.

4. -Figure 2: Do we really need it? Can't we just discuss in the main text? This looks like a rather standard research design. In the text, provide a stronger justification for the countries object of assessment.

  • This image accompanied the readers and researcher throughout the investigation procedure. So it is a big picture of the research for a more convenient procedure follow up.

5. -Tables 1-8: add one column where you list the references in support of each claim made. 

  • These tables were derived from the descriptions of each country's community energy business models in each case study, which were fully referenced in the text. So, in response to your comment, this line” Table --- provides a summary of the CEBMs in ----” is put at the end of each case study paragraph and above each table.

6. -In the conclusions, you use the verb "surveyed". Was a survey conducted? No, so please amend.

  • It has been corrected.

7. -In the conclusions, can we have stronger recommendations, or is not possible based on the research conducted? 

For instance:

In developing countries, community energy projects demand a more accountable and democratic political structure that give community members ownership and participation in policy and decision-making. in this regard, it is vital to transform centralized energy systems into more dispersed ones, as well as a transformation of current institutions and regulations.

 A supportive legal framework in developing countries should be based on proven and tested laws in developed countries, but should be appropriately tailored to differing specific situations and regions.

How is this adding value to previous research? 

  • I added more recommendations in the conclusion part.

Overall, community energy projects necessitate a more accountable and democratic political structure that give community members ownership and participation in policy and decision-making.

 In this regard, in developing countries it is necessary to transform the centralized energy systems into more decentralized ones, as well as a transformation of current institutions and policies. Particularly, a supportive legal framework in developing countries should be based on proven and tested laws in developed countries, but should be appropriately tailored to differing specific situations and regions.

Furthermore, smart energy systems and innovative CEBMs developed and proven in developed countries appear to be a viable option for developing countries. These technological advances may alter the role of individuals in energy systems and require higher levels of  awareness, passion, and participation in society.

Round 3

Reviewer 4 Report

I believe the paper can be now considered for publication.